# NIR light-activated nanocomposites combat biofilm formation and enhance antibacterial efficacy for improved wound healing
Irfan Ullah[1,6], Shahin Shah Khan[1,6], Waqar Ahmad[1], Luo Liu[1], Ahmed Rady[2], Badr Aldahmash[2], Yingjie Yu [3] ✉, Jian Wang [4] ✉ & Yushu Wang[5] ✉

Nanoparticle-based therapies are emerging as a pivotal frontier in biomedical research, showing their potential in combating infections and facilitating wound recovery. Herein, selenium-tellurium dopped copper oxide nanoparticles (SeTe-CuO NPs) with dual photodynamic and photothermal properties were synthesized, presenting an efficient strategy for combating bacterial infections. In vitro evaluations revealed robust antibacterial activity of SeTe-CuO NPs, achieving up to 99% eradication of bacteria and significant biofilm inhibition upon near-infrared (NIR) irradiation. Moreover, in vivo studies demonstrated accelerated wound closure upon treatment with NIR-activated SeTe-CuO NPs, demonstrating their efficacy in promoting wound healing. Furthermore, SeTe-CuO NPs exhibited rapid bacterial clearance within wounds, offering a promising solution for wound care. Overall, this versatile platform holds great promise for combating multidrug-resistant bacteria and advancing therapeutic interventions in wound management.

Pathogenic microbes have emerged as a significant threat to human health[1,2]. Antibiotics are effective against both fungal and bacterial diseases and infections[3,4]. However, the prevalence of antibiotic-resistant bacteria is on the rise, particularly in severe infections, making treatment increasingly challenging. Furthermore, the overuse of antibiotics in clinics leads to the emergence of antibiotic-resistant bacteria[5,6]. Therefore, developing antibacterial materials is of great significance in biomedical research.

The combination of photothermal therapy (PTT) and photodynamic therapy (PDT) is an efficient strategy against multi-drug resistant bacteria[7-10]. Metal-based nanoparticles (NPs), such as copper (Cu), silver (Ag), gold (Au), exhibit unique optical characteristics, which are promising PDT and PTT agents[11-13]. These NPs can strongly interact with adjacent molecules, including the transfer of electrons from excited molecules to metal NPs and the transfer of energy from metal NPs to nearby molecules[14].

Therefore, metal-based NPs can be used to enhance the efficiency of photosensitizers in PDT and PTT[15-17].

Cu exhibits potent bioactivity that can be utilized to eradicate bacteria[18-21]. Furthermore, high ROS generation from intracellular excess hydrogen peroxide ($H_2O_2$) can be catalyzed by Cu *via* a fenton like reaction[22], hence inducing bacterial cell death[23]. Applying Cu in wound dressing materials can prevent infections[24]. Owing to the low in vivo toxicity, Cu can be used for the surface coating of implants[25,26]. The antimicrobial capability of CuO NPs has already been demonstrated and exploited[27-29]. CuO NPs were effective against pathogenic bacteria involved in hospital acquired infections. However, high concentrations of CuO NPs are required for bactericidal effect. To address this issue, surface modification is employed to improve the antibacterial performance of CuO based materials[30].

[1]College of Life Science and Technology, Beijing University of Chemical Technology, No. 15 East Road of North Third Ring Road, Chao Yang District, Beijing 100029, China. [2]Department of Zoology, College of Science, King Saud University, P.O. Box 2455, Riyadh 11451, Saudi Arabia. [3]State Key Laboratory of Organic-Inorganic Composites, Beijing University of Chemical Technology, No. 15 East Road of North Third Ring Road, Chao Yang District, Beijing 100029, China. [4]Department of Head and Neck Surgery, National Cancer Center/National Clinical Research Center for Cancer/Cancer Hospital, Chinese Academy of Medical Sciences and Peking Union Medical College, Beijing 100021, China. [5]The People's Hospital of Gaozhou, National Drug Clinical Trial Institution, Gaozhou City 525200, China. [6]These authors contributed equally: Irfan Ullah, Shahin Shah Khan. ✉e-mail: yuyingjie@mail.buct.edu.cn; wangjianpumc@126.com; wysmjeda@gmail.com

Herein, this study reports a one-pot synthesis of SeTe-CuO NPs for dual-mode therapy that combines photothermal and photodynamic properties for tackling bacterial infections. This synergistic effect significantly enhances antibacterial efficacy (Fig. 1). Notably, SeTe-CuO NPs exhibit significant capabilities in eradicating both *Staphylococcus aureus* (*S. aureus*) and *Escherichia coli* (*E. coli*) *via* the synergistic effect of photothermal and photodynamic effect, effectively targeting biofilm-associated pathogens. Beyond their antibacterial and anti-biofilm properties, SeTe-CuO NPs demonstrate promising potential in angiogenesis, hence promoting wound healing. Overall, this study underscores the versatility of SeTe-CuO NPs as a promising platform for combating bacterial infections.

## Results and discussions
### Synthesis and characterization of SeTe-CuO NPs
SeTe-CuO NPs were synthesized using a one-pot synthesis method. In Fig. 2A, transmission electron microscopy (TEM) images show the morphology of various NPs (Fig. 2A). Furthermore, energy dispersive spectroscopy (EDS) displays the metalloid amalgamation of SeTe-CuO NPs, confirmed by high resolution mapping (Supplementary Fig. 1). As depicted in Fig. 2B, the obtained EDS spectrum shows the elements of Se, Te, O, and Cu, which demonstrates the successful doping of CuO NPs on the surface of Se-Te NPs. Notably, the crystallinity of the Se-Te NPs was maintained. As indicated by the X-ray diffraction (XRD) pattern, the respective diffraction bands closely matches the diffraction profile of the chalcogenide family (Fig. 2C)[31]. The prominent diffraction peak at 28.77° indicates that cetyltrimethylammonium bromide (CTAB) directs the formation of Se-Te NPs on the (101) lattice plane[32,33]. Furthermore, the CuO NPs were immobilized on the surface of the Se-Te NPs. The diffraction bands of the CuO NPs appeared at 35.32° and 38.54° (Fig. 2C)[34].

The x-ray photoelectron spectroscopy (XPS) survey spectra of SeTe-CuO NPs are presented in Fig. 2D. The elemental composition of the NPs is confirmed by Cu 2p, Se 3d, Te 3d, O1s, C1s, and N1s (Supplementary Fig. 2). Cu 2p is deconvoluted into three peaks at binding energies of 932.64 for Cu for $2p_{3/2}$, 941.8 eV as satellite peak and 953.7 eV for Cu $2p_{1/2}$, which corresponds to the presence of Cu as I and II oxides (Fig. 2E)[35]. Two different bands were observed for Se 3d at binding energies of 55.1 eV and 58.6 eV. Similarly, four different peaks were observed for Te 3d at 573.5, 575.9, 583.9, and 586.5 eV. The peaks observed at 573.5 and 583.9 eV represent $3d_{5/2}$ and $3d_{3/2}$ of Te(0)3d, while the other two peaks correspond to Te(IV)3d. The

coexistence of the two forms indicates that Te exists in both the elemental and oxide forms.

Different functional groups responsible for stabilizing and reducing CuO NPs, as well as those in the NPs, were analyzed by Fourier transform infrared (FTIR) spectroscopy. The absorption peak at 3442 cm$^{-1}$ corresponds to the OH stretch in CTAB-mediated SeTe-CuO NPs (Fig. 2F). Furthermore, the Raman spectra of SeTe-CuO NPs and CTAB mediated Se-Te NPs were investigated at room temperature (Fig. 2G). Raman spectroscopy was employed to elucidate the structural relationship between SeTe-CuO NPs and CTAB-stabilized Se-Te NPs, as depicted in (Fig. 2G). The prominent peak observed at approximately 130 cm$^{-1}$ corresponds to the characteristic vibrational modes of Se-Te NPs, which shifted slightly upon the introduction of CuO NPs due to changes in the local electronic environment and strain effects within the crystal lattice of Se-Te NPs. This shift in SeTe-CuO NPs indicates a change in peak intensity suggesting possible changes in the chemical bonding or structural arrangement of the Se-Te NPs lattice. This change in peak intensity observed when CuO NPs are introduced, which might significantly enhance the Raman signal by increasing the electromagnetic field around the NPs. These findings are critical as they suggest that CuO NPs incorporation alters the properties of Se-Te NPs that might also potentially improves their applicability in various applications. Overall, these results confirm the formation of SeTe-CuO NPs.

### Evaluation of photodynamic and photothermal properties
The photodynamic properties of SeTe-CuO NPs were investigated using electron paramagnetic resonance (EPR). The sample was irradiated with 808 nm laser (0.8 W cm$^{-2}$) for five minutes. Then, 5,5 – dimethyl -1-pyrroline-N-oxide (DMPO) and 2,2,6,6-tetramethylpiperidine (TEMP) were used to detect hydroxyl and singlet oxygen radicals, respectively. A 2,2,6,6-tetramethyl-1-piperidinyloxy (TEMPO) adduct was produced by the reaction of $^1O_2$ with TEMP (singlet oxygen spin trapping reagent). The EPR spectrum indicates ROS generation of SeTe-CuO NPs upon NIR laser irradiation (Fig. 3A). Electron excitation in samples irradiated with NIR laser could accelerate ROS production from dissolved oxygen. Additionally, EPR analysis shows a pronounced signal for hydroxyl radical generation from the DMPO-OH adduct in SeTe-CuO NPs. Conversely, only a negligible EPR signal was observed in Se-Te NPs upon the NIR irradiation (Fig. 3B).

We continued to investigate the photothermal performance of Se-Te NPs and SeTe-CuO NPs. Upon irradiation with NIR 808 nm laser,

**Fig. 1 | Schematic illustration showing the synthesis of NIR light-activated SeTe-CuO NPs.** The anti-bacteria and wound healing effect were achieved using PDT and PTT of SeTe-CuO NPs.

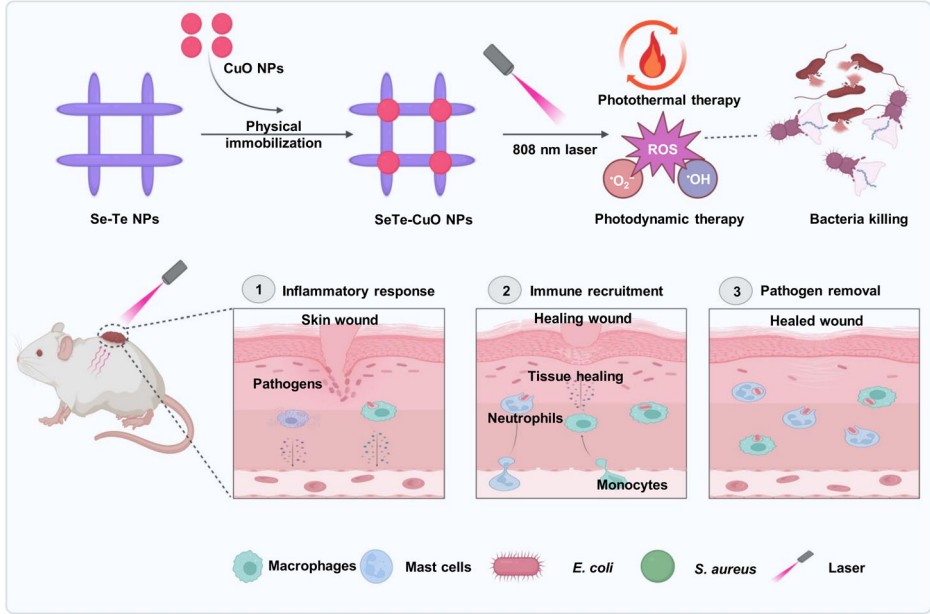

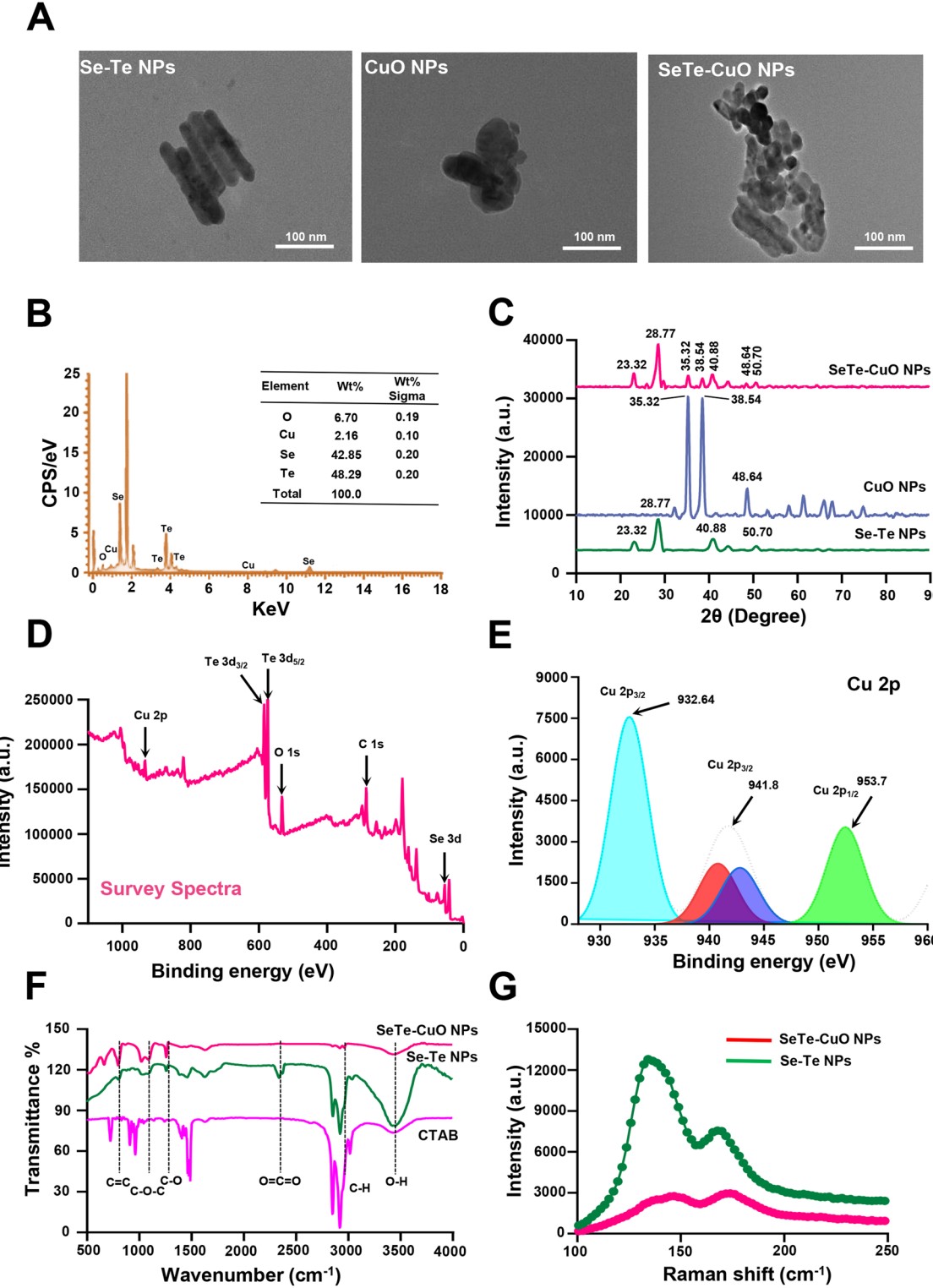

**Fig. 2 | Characterization of SeTe-CuO NPs. A** Transmission electron microscopy (TEM) images of Se-Te NPs, CuO NPs, and SeTe-CuO NPs. **B** Energy dispersive spectroscopy (EDS) elemental spectrum and quantification results of SeTe-CuO NPs. **C** X-ray diffraction (XRD) patterns of Se-Te NPs, CuO NPs, and SeTe-CuO NPs. **D** X-ray photoelectron spectroscopy (XPS) survey spectra of SeTe-CuO NPs. **E** Deconvoluted spectra of Cu 2p of SeTe-CuO NPs. **F** Fourier transform infrared spectroscopy (FTIR) spectra of CTAB, Se-Te NPs, and SeTe-CuO NPs. **G** Raman spectra of Se-Te NPs and SeTe-CuO NPs.

the temperature of the SeTe-CuO NPs reached 60.2 °C, while the temperature of Se-Te NPs reached only 48 °C (Fig. 3C). An infrared thermal camera was used to monitor the temperature change. As shown in Fig. 3D, to test the photostability, a suspension of SeTe-CuO NPs was irradiated with NIR 808 nm laser for 5 min and followed by cooling period after switching off the laser (Supplementary Fig. 3). There was only a negligible photothermal transformation effect as well as deterioration, confirming the excellent photostability of SeTe-CuO NPs[36]. The conversion rate of light to heat was calculated according to a previously published method[37].

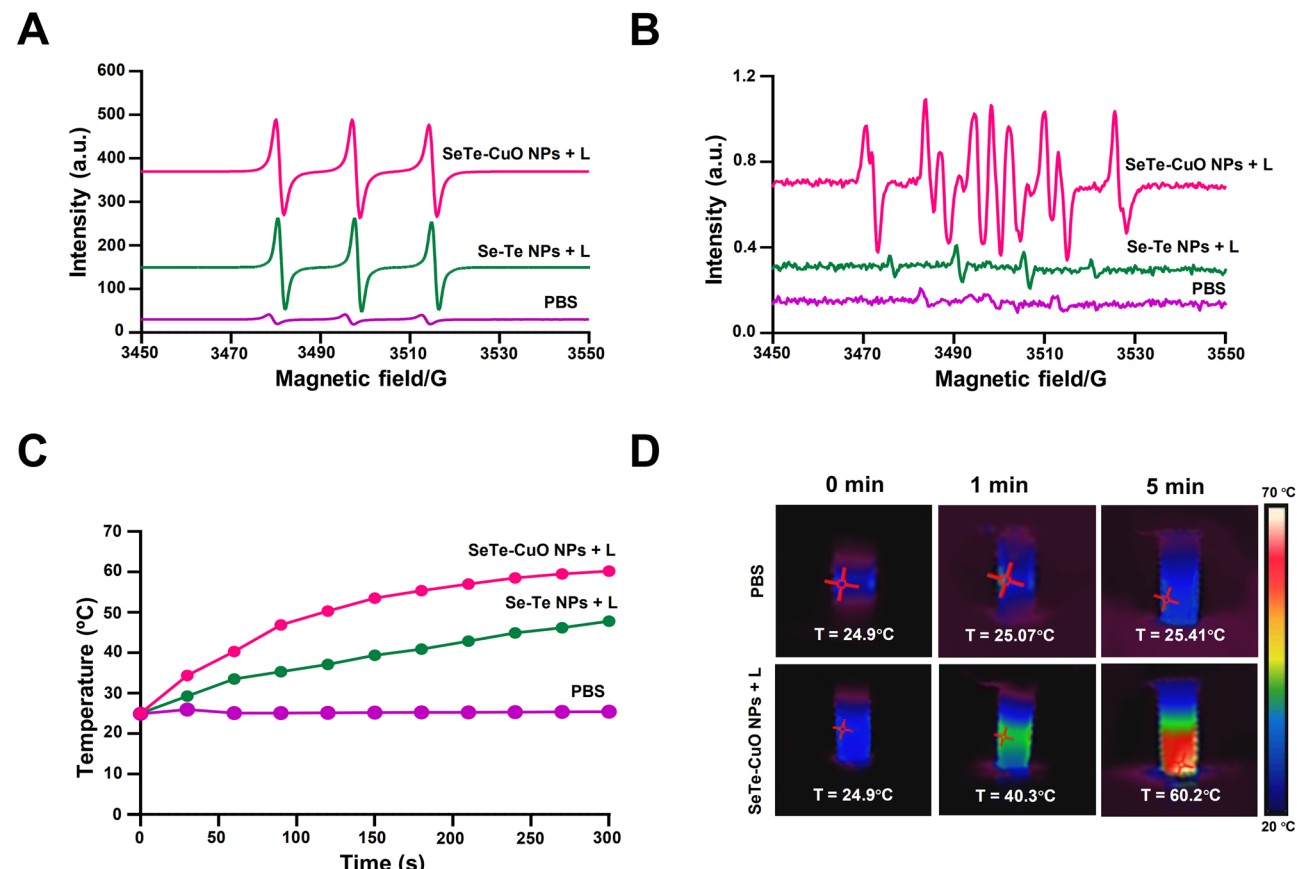

**Fig. 3 | Evaluation of photodynamic and photothermal properties.** Electron paramagnetic resonance (EPR) shows the characterization of (**A**) singlet oxygen spectra and (**B**) hydroxyl radical spectra. **C** Temperature profiles of various NPs solution after 808 nm laser irradiation. **D** Thermal images of NPs solution upon 808 nm laser irradiation.

## Evaluation of antibacterial and antibiofilm activity

The antibacterial activity of SeTe-CuO NPs was assessed using the agar well diffusion method against *E. coli* and *S. aureus*. The NPs were added into the diffused well and incubated for 24 h, followed by measuring the inhibition zone in mm. The synthesized NPs demonstrated higher antibacterial activity against both bacterial strains as expected. The recorded diameter of the inhibition zone of SeTe-CuO NPs against *E. coli* was $35 \pm 2$ mm and against *S. aureus*, it was $30.66 \pm 2$ mm (Fig. 4A, Supplementary Fig. 4). The results show that SeTe-CuO NPs possess significant antibacterial activity against both *E. coli* and *S. aureus* strains as compared to CuO NPs and Se-Te NPs.

Furthermore, antibiofilm activity of SeTe-CuO NPs was evaluated against *S. aureus* and *E. coli*. Biofilm formation in the presence of NPs was inhibited in a dose-dependent manner (Fig. 4B, C). The biofilm formation was assessed by the mass of biofilm which was stained with crystal violet. Based on these results, at $100 \, \mu g \, mL^{-1}$ there was about an 80% reduction in biofilm for *E. coli* while about 70% of the biofilm was inhibited in *S. aureus*[38].

To evaluate the growth patterns of bacteria, bacterial cultures with an $OD_{600}$ of 0.4 were treated with different concentrations of NPs and incubated for 10 h at 37 °C. The growth of bacteria was inhibited under various concentrations, while the untreated bacteria grew naturally and reached a stationary phase after 24 h of incubation. These results clearly indicate that the antibacterial activity of the synthesized NPs increases with an increase of concentration[39]. Moreover, the antibacterial activity of the NPs was also evaluated using the agar plate method (Fig. 4D). Both with and without NIR laser irradiation, the synthesized NPs exhibited excellent antibacterial activities against both *E. coli* and *S. aureus*[40–42].

Furthermore, the effect of SeTe-CuO NPs on bacteria was studied through membrane disruption. Scanning electron microscopy (SEM) images show that untreated *E. coli* exhibited a rod shape and *S. aureus*

exhibited a typical spherical shape, both with smooth and intact surfaces. However, bacteria treated with SeTe-CuO NPs underwent significant membrane damage (Fig. 4E)[14,43,44]. Subsequently, the live/dead assay was used to assess the antibacterial activity of SeTe-CuO NPs (Fig. 4F)[45,46]. Bacteria stained with DAPI emitted blue fluorescence, indicating the presence of live bacteria. Confocal laser scanning microscopy (CLSM) images show that the SeTe-CuO NPs treatment group exhibit an increase in red fluorescence emission, indicative of a large number of dead bacteria. Furthermore, the ratio of live/dead bacteria was significantly higher in the group treated with SeTe-CuO NPs + L, which confirms that the membrane of bacteria was highly damaged. When stained with PI, or DAPI + PI, the bacteria emitted red fluorescence[47]. Overall, the above results collectively demonstrate that the designed NPs possess strong antibacterial activity due to the synergistic effect SeTe-CuO NPs upon NIR laser irradiation.

## Antibacterial mechanism

The bactericidal activity is attributed to the release of metallic ions and the close interaction of the NPs with the bacterial membrane. CuO NPs exhibit enhanced microbicidal activity against various pathogenic microbes[29]. Cu NPs accumulate on the surface of bacterial cells, decreasing the transmembrane electrochemical potential, which disrupts the membrane integrity. The accumulation of CuO NPs creates holes on the surface of bacterial cells leading to the leakage of the intracellular components and facilitating the penetration of CuO NPs into the cell[34]. The smaller size of NPs makes it feasible to penetrate the cell wall and release metal ions. ROS generation is accelerated by $Cu^{2+}$ ions, causing an imbalance in ROS production. The resultant hydroxyl changes inflict serious damage to cells, such as protein and DNA damage. Consequently, the ROS production by SeTe-CuO NPs and SeTe-CuO NPs + L was evaluated (Fig. 4G), and their corresponding fluorescence intensities are quantified in supplementary

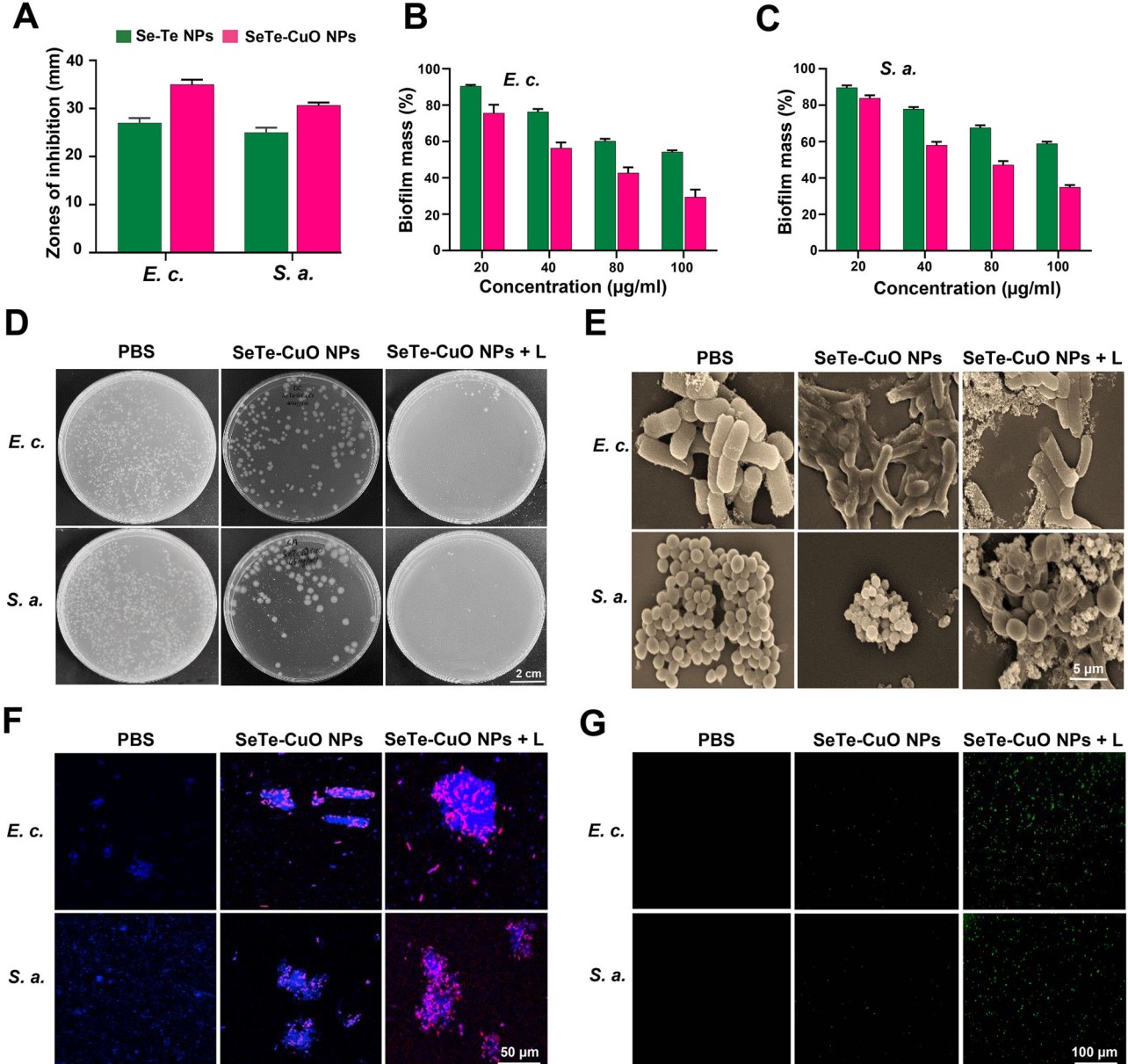

**Fig. 4 | Evaluation for in vitro antibacterial effect of SeTe-CuO NPs. A** Zone of inhibitions of Se-Te NPs and SeTe-CuO NPs. **B, C** Antibiofilm activity of Se-Te NPs and SeTe-CuO NPs against *E. c.* and *S. a.* **D** Antibacterial activity using agar plate method in the absence and presence of 808 nm laser irradiation. **E** SEM images of bacteria treated with various NPs. **F** CLSM images of live/dead staining. **G** CLSM images of intracellular ROS generation. In (**A, B, C**), the graphs represent mean values and the error bars correspond to standard deviations, *n* = 3.

Fig. 5. The fluorescence increases gradually with the order of SeTe-CuO NPs and SeTe-CuO NPs + L, confirming the increased ROS production.

**In vivo antibacterial activity and wound healing**

To assess the cytotoxicity of SeTe-CuO NPs, CCK-8 cell viability assay was employed. As demonstrated in Fig. 6B, with the increasing concentration of NPs, there was a negligible influence on the viability of L929 cells. The cell viability remained over 95% even when the concentration of SeTe-CuO NPs increased to 100 µg mL$^{-1}$, indicating excellent biocompatibility of the synthesized NPs. Additionally, the hemolysis experiment was also conducted to investigate the blood biocompatibility of SeTe-CuO NPs by incubating with mice blood cells. As depicted in Supplementary Fig. 7, the NPs demonstrate low hemolysis. At a higher concentration of 100 µg mL$^{-1}$, the NPs showed about 5% of hemolysis, while cells treated with water were used as positive control. These results show that SeTe-CuO NPs did not cause hemolysis and possess a high level of biocompatibility[48].

Having confirmed the excellent in vitro antibacterial activity, the in vivo antibacterial activity of the synthesized NPs was investigated using a bacteria infected wound healing therapy model with BALB/c mice (Fig. 5A)[49]. The mice were randomly divided into 4 groups for evaluation. Subsequently, an equal sized wound was constructed on the back of the mice using a hole puncher, and 10 µL bacteria suspension with an OD$_{600}$ of 0.4 was injected. After 24 h of infection, the wounds were treated with Se-Te NPs, SeTe-CuO NPs, and SeTe-CuO NPs + L. After 12 days of treatment, the wound size for the mice treated with SeTe-CuO NPs + L sharply reduced with a recovery rate of 93% in comparison to that of Se-Te NPs and SeTe-CuO NPs (Fig. 5B, C). The scar in the SeTe-CuO NPs treatment group significantly reduced and disappeared after day 12. Overall, the results clearly demonstrate that the synergistic effect of SeTe-CuO NPs with NIR 808 nm laser enhanced the antibacterial activity against *E. coli*, significantly improving wound healing compared to other treatment groups. Moreover, there was no significant change in the body

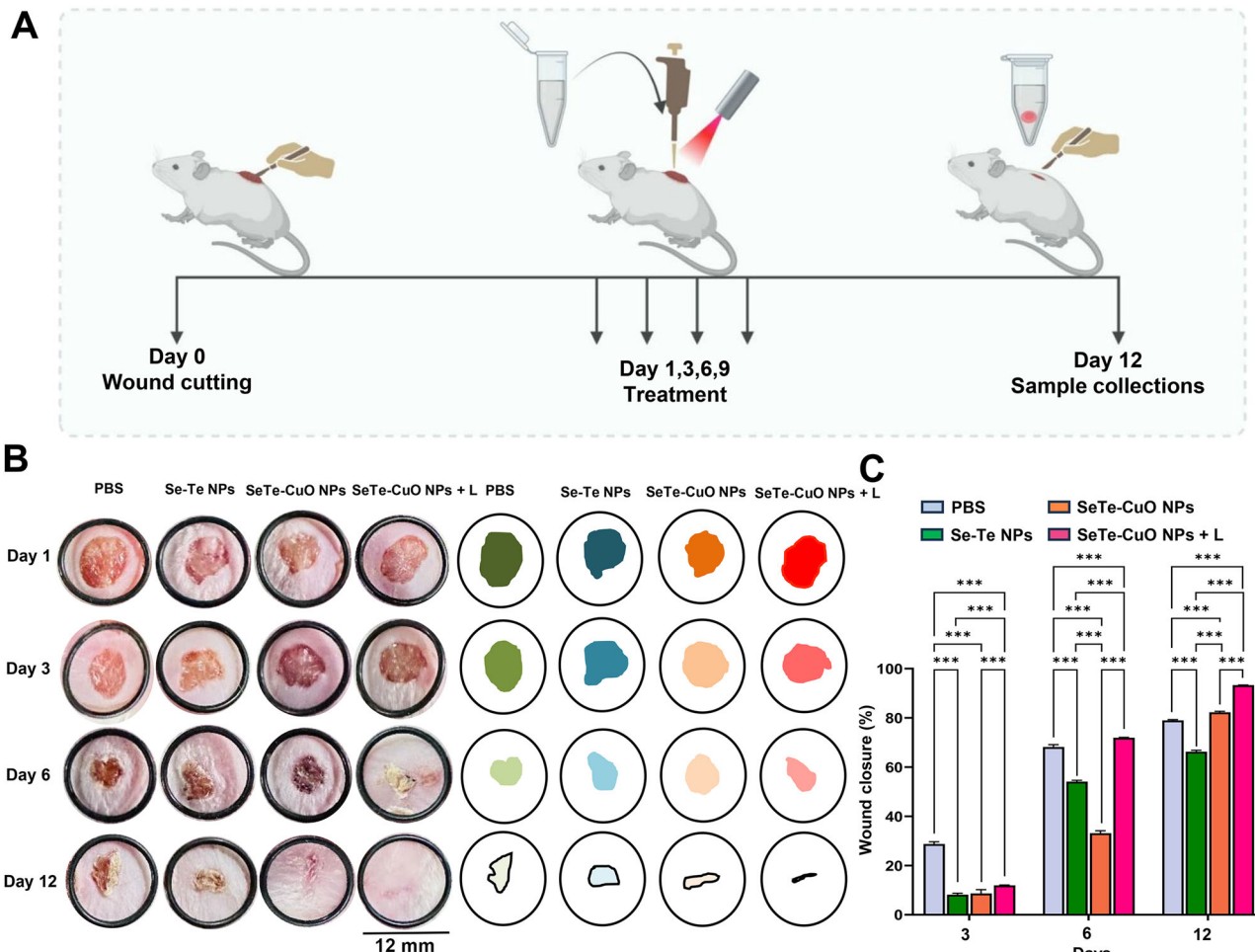

**Fig. 5 | Evaluation of wound healing. A** Schematic illustration of in vivo study. **B** Representative photographs of the wound healing images and their corresponding wound healing area. **C** Wound closure percentages of different treatment groups on different days, and the data represent mean values and the error bars correspond to standard deviations, Student's $t$ test was used for calculating $p$ value, *** represents $p \leq 0.05$, $n = 3$.

weight of any treatment groups, indicating the biosafety of the NPs (Supplementary Fig. 6).

### Histopathology and giemsa staining

Hematoxylin and Eosin (H&E) staining was performed to investigate the treatment of the bacterial infected wounds (Fig. 6A). The presence of large number of neutrophils in the PBS treatment group indicates the gathering of inflammatory cells due to infection. In contrast, the Se-Te NPs, SeTe-CuO NPs, and SeTe-CuO NPs + L treatment groups had an intact epidermal layer and collagen fibers, as well as a reduced number of inflammatory cells, which indicates improved reepithelization and enhanced wound therapy under NIR laser irradiation. Furthermore, all treatment groups exhibited elongated epithelial cells and fibroblasts on day 12. The development of hair follicles and thick granulation tissue was observed in SeTe-CuO NPs + L treatment group, demonstrating an enhanced wound healing efficiency of the synthesized NPs. The results obtained demonstrate that the therapeutic efficiency of SeTe-CuO NPs was enhanced with the application of NIR laser irradiation.

The infection of the wound is able to delay the wound healing process[50]. Therefore, Giemsa staining of the excised tissue was performed to assess the bacterial infection on days 6 and 12. In both the PBS and SeTe-CuO NPs treatment groups, a large number of bacteria were observed in the wound on day 6 (Fig. 6C). However, a lower number of bacteria or infections was observed in the SeTe-CuO NPs + L treatment group. The number of bacterial residues or infections was not detected in the SeTe-CuO NPs + L

treatment group on day 12. This was also evidenced by the bacteria grown on agar plates from the healing skin (Fig. 6D). This showed that the extent of bacterial killing could be due to the intrinsic bactericidal ability of the Se-Te NPs and CuO NPs and indicates that the activity of the SeTe-CuO NPs was enhanced with the application of NIR laser irradiation. The data obtained shows that the infected wound healing ability of the NPs was improved with NIR laser irradiation. Based on these results, SeTe-CuO NPs with NIR laser irradiation exhibit satisfactory antibacterial performance, confirming the in vitro antibacterial activity of the NPs.

### Conclusions

In this study, a facile one-pot synthesis method was used for the synthesis of composite NPs for antibacterial activity against both gram-negative and gram-positive bacteria. The therapeutic efficacy of the synthesized NPs enhanced with NIR laser irradiation. In cytotoxicity investigation, SeTe-CuO NPs were found to have good biocompatibility, suggesting further exploration as a potential treatment against drug-resistant bacteria. Furthermore, the mechanism of action exploration of the bactericidal activity shows that the synthesized NPs destroy cell integrity, produces ROS, and damages the bacterial cell membrane. In vivo study showed enhanced wound healing due to the remarkable deposition of collagen and reepithelialization characteristics. The synergistic activity of the dual or multiple NPs could significantly reduce the dosage requirements for conventional antibiotics, hence providing a versatile platform for the treatment of the drug-resistant bacteria.

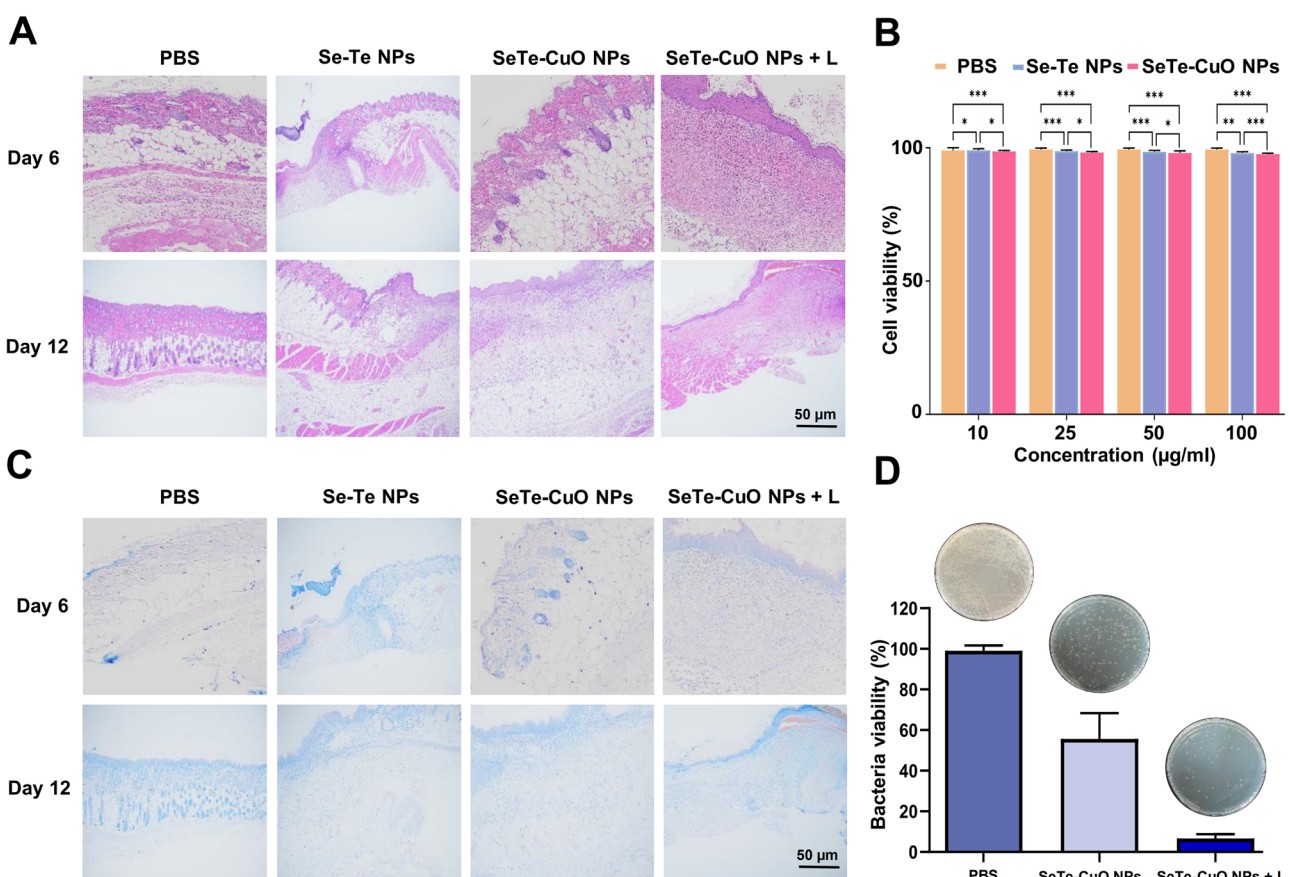

**Fig. 6 | Cytotoxicity and histological study of wound tissues. A** Hematoxylin and eosin (H&E) staining images of wound regeneration. **B** Cytotoxicity assessment of various NPs using CCK-8 assay. **C** Giemsa staining of wound tissues.

**D** Representative images and quantitative results of the bacteria from wounds of different treatment groups. In (**B**, **D**), the graphs represent mean values and the error bars correspond to standard deviations, $n = 3$.

## Materials and methods

### Materials
The chemicals used in this study were purchased from Sigma Aldrich, St. Louis, MO, USA, i.e., sodium selenite ($Na_2SeO_3$), telluric acid ($H_6TeO_6$), cetyltrimethylammonium bromide (CTAB), hydrazine, ascorbic acid, 2,2,6,6-tetramethylpiperidine (TEMP), 4,6-diamidino-2- pheny-lindole (DAPI), 5,5 – dimethyl -1-pyrroline-N-oxide (DMPO) and 2,7 – Dichlorofluorescein diacetate (DCFH-DA). The bacterial strains, *E. coli* (ATCC8739) and *S. aureus* (ATCC6538), were acquired from China General Microbiological Culture Collection Centre, Chinese Academy of Sciences, Beijing, China.

### Synthesis of SeTe-CuO NPs
The synthesis of Se-Te NPs was carried out using two different reducing agents i.e., ascorbic acid and hydrazine. Briefly, telluric acid (20 mM) and sodium selenite (20 mM) were prepared in the presence of CTAB (2 mg mL$^{-1}$) with the final volume adjusted to 100 mL. The solution was sonicated for 30 min, followed by stirring in an oil bath at 250 rpm, at 95 °C for 3 h. Then a mixture of reducing agents (hydrazine (500 µL) and ascorbic acid (100 mg mL$^{-1}$)) prepared in 10 mL of dH$_2$O, was added slowly to the reaction and continued for 30 min at 95 °C. An abrupt change in color from colorless to deep gray was observed. Afterward, the product was collected and purified using centrifugation and dried at 60 °C overnight. Then, the synthesized Se-Te NPs (0.3 g) were dissolved in 30 mL of water and sonicated for 30 min, followed by 2 mM of CuO NPs were dissolved in 10 mL of water using sonication for 30 min and were mixed dropwise with the as synthesized Se-Te NPs solution. The solution was vigorously stirred for 2 h at 75 °C. Then, 3 mL of hydrazine was added to reduce the free materials in the solution. Finally, SeTe-CuO NPs were collected and purified using centrifugation followed by drying overnight at 60 °C.

### Anti-bacterial and antibiofilm assessments
The assessments of the antibacterial activities of SeTe-CuO NPs were performed using two different bacterial strains, *E. coli* and *S. aureus*. For culturing bacteria, Luria Bertani (LB) broth media was used and different concentrations of SeTe-CuO NPs were used i.e., 12, 24, 48, and 96 µg mL$^{-1}$. The synthesized NPs were mixed with bacterial cells and the mixture was irradiated with an NIR laser with a power density of 0.8 W cm$^{-2}$ for 5 min. Followed by incubation at 37 °C for 2 h, and then a 100 µL of aliquot was plated and cultured overnight at 37 °C. For assessment of bacterial growth patterns under the NPs influence, bacterial culture with an optical density (OD) of 0.4 were treated with different dosages of NPs, and the culture was grown further for 10 h measuring OD$_{600}$ every hour. Moreover, for the antibiofilm activity of NPs, a bacterial cell culture of 0.025 OD$_{600}$ was added to 96 well plate and grown at 37 °C for 24 h. The media was removed by inverting the plate after biofilm formed on the walls of the plates, and the planktonic cells were removed by washing the biofilm with PBS. Followed by the addition of different concentrations of SeTe-CuO NPs (20, 40, 80, and 100 µg mL$^{-1}$) prepared in PBS, to the wells, PBS treated group was taken as negative control. The biofilm was fixed by adding crystal violet (0.5%) and methanol for 15 min. Biofilm was washed using sterile PBS, and the crystal violet was dissolved using acetic acid 33% (V/V), with gentle shaking for 10 min. For biofilm quantification, 96 well microplate reader was used to measure absorbance at 590 nm.

### Live/dead assay of bacterial cells

For the assessment of live/dead bacteria, fluorescent based live/dead assay method was applied. Briefly, 1500 µL of bacteria were washed with PBS (pH 7.4), and treated with 48 µg mL$^{-1}$ SeTe-CuO NPs, followed by irradiation with NIR laser (808 nm) with a power density of 0.8 W cm$^{-2}$ for 10 min. Afterward, the mixture was cultured for one hour, and then propidium iodide (PI 50 µL, 30 µM) a fluorescent dye was added, and incubated for 15 more minutes. Followed by washing with PBS three times to remove extracellular dye. For observation of the samples, a confocal laser scanning microscopy (CLSM) (Leica DMI, 4000 B, Danaher, Duesseldorf, Germany) was used to capture the photographs.

### Intracellular ROS detection of bacteria

The production of reactive oxygen species (ROS) was analyzed according to the method previously reported[51], with small changes. DCFH-DA (10 µM), an intracellular ROS detection dye was added to 20 mL (0.9%) NaCl solution and incubated in the dark with $1 \times 10^6$ CFU mL$^{-1}$ bacterial cells for 30 min. The treatment of bacterial cells was performed as following groups: PBS, SeTe-CuO NPs, SeTe-CuO NPs + L. Treatment was applied for 5 min, followed by incubation at 37 °C for 4 h. The fluorescence (λex = 485 nm, λem = 520 nm) was measured using a fluorescence spectrophotometer.

### SEM characterization of bacteria

Overnight grown bacterial cell culture was collected, washed three times, and incubated with 48 µg mL$^{-1}$ of SeTe-CuO NPs for one hour. All samples were treated with NIR laser for 10 min, centrifugation was performed at 5000 rpm, and washed with PBS. Followed by overnight fixing of the cells with 2.5% glutaraldehyde solution at 4 °C, subsequently, different concentrations of ethanol were used for dehydration. A scanning electron microscope (SEM, Hitachi SU, 8080, Tokyo, Japan) was used for morphological assessment.

### Biocompatibility evaluation

The biological toxicity evaluation of SeTe-CuO NPs was performed using CCK-8 assay kit, at concentrations of 10, 25, 50, and 100 µg mL$^{-1}$ with L929 mouse fibroblast cells. After 24 h incubation, the cytotoxicity of NPs was tested using the CCK-8 cell viability assay kit, following the manufacturer's protocols. Additionally, a hemolysis test was also performed to evaluate blood hemolysis ratio of the NPs. Fresh mice blood was collected, and centrifugation was performed at 1500 rpm for 15 min to concentrate red blood cells and washed three times with normal saline. The concentrated red blood cells were diluted to 5% and different treatment groups were prepared; i.e., water, PBS, and different concentrations of SeTe-CuO NPs (25, 50, and 100 µg mL$^{-1}$) and incubated for 3 h at 37 °C, followed by centrifugation for 5 min at 11,000 rpm. The supernatant was collected and absorbance was checked at 540 nm. The percent rate of hemolysis was calculated for SeTe-CuO NPs with water treatment group as control.

### In vivo wound healing analysis

All animal experiments reported in this study were conducted in accordance with the guidelines (ZYZY202209005S) assessed and approved by the institutional animal care and use committee of Sino Research (Beijing) Biotechnology *Co., Ltd.* (China).

Wound healing and in vivo antibacterial tests were carried out on female BALB/c (5-6 weeks old,18 ± 2 g). All mice were subjected to wound formation on the skin according to the reported method[52]. For the aseptic server, 7% sodium sulfide was used to remove dorsal hairs, and then 75% ethanol was used for disinfection. A round wound cut of 6 mm diameter was created on the back of each mouse using a hole puncher. After 24 h, 20 µL of the synthesized NPs were applied to the wounds. The treatment of the wounds was performed every second day. The wounds of every mouse were imaged with a camera. The area of the wound was calculated with ImageJ software. The percent healing rate of the wounds was calculated using the following formula:

$$\text{Wound healing rate}\,(\%) = \frac{\text{Area Day 0} - \text{Area Day X}}{\text{Area Day 0}} \times 100$$

Area Day 0 and Area Day X were the areas of the wound on Day 0 and Day X respectively. The mice were sacrificed and the wound tissues were collected for histological analysis on days 6 and 12. For wound histological analysis, the collected wound tissues from each treatment group were fixed in formalin (10%) and stained with hematoxylin-eosin (H&E).

### Reporting summary

Further information on research design is available in the Nature Portfolio Reporting Summary linked to this article.

## Data availability

The data that support the findings of this study are available from the corresponding author upon reasonable request. Supplementary information data includes EDS pictures of SeTe-CuO NPs (Supplementary Fig. 1), XPS elemental spectra of SeTe-CuO NPs. (Supplementary Fig. 2), Single and five consecutive cycles of SeTe-CuO NPs + L with 5 min NIR laser irradiation ON and then OFF (Supplementary Fig. 3), Zone of inhibitions of SeTe-CuO NPs against *S. aureus* and *E. coli* (Supplementary Fig. 4), Fluorescence intensities (FL) of intracellular ROS generation of *E. coli* and *S. aureus* after treatment with SeTe-CuO NPs and SeTe-CuO NPs + L (Supplementary Fig. 5), Body weight of mice after treatment with PBS, SeTe-CuO NPs and SeTe-CuO NPs + L (Supplementary Fig. 6), and Hemolysis ratio of the mice blood samples after treatment with Se-Te NPs and SeTe-CuO NPs (Supplementary Fig. 7). All the numerical data belongs to figures in the main manuscript, and supplementary information is present in supplementary data 1.

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

## Acknowledgements

The authors gratefully acknowledge the financial support of the National Natural Science Foundation of China project (52103084), China Postdoctoral Science Foundation (2021M701591), and Research Supporting Project No. (RSP2024R214) King Saud University, Riyadh, Saudi Arabia.

## Author contributions

Irfan Ullah: Experimental work, writing original draft. Shahin Shah Khan: Conceptualization, experimental work, data analysis, and reviewing the draft. Waqar Ahmad: Formal analysis, Luo Liu: Conceptualization, draft modification. Ahmed Rady: Conceptualization, Badr Aldahmash: Conceptualization, Yingjie Yu: Conceptualization, supervision, and reviewing the draft, Jian Wang: Reviewing the draft, Yushu Wang: Conceptualization, reviewing the draft.

## Competing interests

The authors declare no competing interests.
