## [Peer Review File · Communications Chemistry]

Reviewers' comments:

Reviewer #1 (Remarks to the Author):

In the manuscript, Ullah and colleagues describe a novel nanoparticle (SeTe-CuO NPs) composed of SeTe and CuO to be used for wound healing. Particularly, SeTe-CuO NPs exhibited excellent photodynamic and photothermal properties, which is conducive to the anti-bacteria performance. Generally, the message of the article is important, novel and within scope for Communications Chemistry. Findings are abundant, technically diverse, and well presented. I am listing a few concerns and making suggestions that should enhance the manuscript. Given the great future importance of these studies, every possible artefact and misinterpretation must be ruled out. In summary, I believe the article by Ullah and colleagues represents a strong candidate for publication in Communications Chemistry upon revision.

1. In Scheme. The labeling for "Laser 808 nm" should be corrected as "808 nm laser".
2. For all the images, the format for x and y axis should be adjusted.
3. In Figure 4F, 4G, 6A and 6C, the scale bar is missing.
4. The photothermal conversion efficiency for SeTe-CuO NPs should be calculated.
5. The language of this work should be polished.

Reviewer #2 (Remarks to the Author):

In this manuscript, the authors synthesized SeTe-CuO NPs with dual photodynamic and photothermal properties. It revealed robust antibacterial activity and significant biofilm inhibition against pathogenic bacteria upon near-infrared (NIR) irradiation. Moreover, SeTe-CuO NPs exhibited rapid bacterial clearance within wounds, offering a promising solution for advanced wound care. Overall, the experiments were well-designed and the results seem solid. I recommend a major revision before acceptance for the publication.

Comments:

1. What is the role of SeTe in SeTe-CuO NPs?

2 In cytotoxicity investigation, it couldn't give the conclusion of "SeTe-CuO NPs were found to have good biocompatibility". The paper only discussed the assessment of SeTe-CuO NPs on live/dead bacteria, the evaluation of SeTe-CuO NPs on normal cells and other biocompatibility experiments also need to be discussed.

3 Compared to other CuO-based materials, what are the advantages of SeTe-CuO NPs in terms of biofilm control and antibacterial efficacy?

4 In Legend 4, the scale in Fig.4E should be added and there is no caption for Fig.4G.

Rebuttal Letter to the Reviewers

Reviewer #1:

In the manuscript, Ullah and colleagues describe a novel nanoparticle (SeTe-CuO NPs) composed of SeTe and CuO to be used for wound healing. Particularly, SeTe-CuO NPs exhibited excellent photodynamic and photothermal properties, which is conducive to the anti-bacteria performance. Generally, the message of the article is important, novel and within scope for Communications Chemistry. Findings are abundant, technically diverse, and well presented. I am listing a few concerns and making suggestions that should enhance the manuscript. Given the great future importance of these studies, every possible artefact and misinterpretation must be ruled out. In summary, I believe the article by Ullah and colleagues represents a strong candidate for publication in Communications Chemistry upon revision.

1. In Scheme. The labeling for “Laser 808 nm” should be corrected as “808 nm laser”.

Response: Thanks for your valuable comment. The label was corrected as suggested. The revised Figure 1 is provided below. All the similar issues have been corrected throughout the paper.

Figure 1. Schematic illustration showing the synthesis of NIR light-activated SeTe-CuO nanoparticles. The anti-bacteria and wound healing effect were achieved using PDT and PTT of SeTe-CuO nanoparticles.

2) For all the images, the format for x and y axis should be adjusted.

Response: Thanks for your comment. The x and y axis of all the figures are adjusted. One of the representative figures is mentioned below.

Figure 2. Characterization of SeTe-CuO NPs. (A) Transmission electron microscopy (TEM) images of Se-Te NPs, CuO NPs, and SeTe-CuO NPs. Scale bar = 100 nm. (B) Energy dispersive spectroscopy (EDS) elemental spectrum and quantification results of SeTe-CuO NPs (C) X-ray diffraction (XRD) patterns of Se-Te NPs, CuO NPs, and SeTe-CuO NPs. (D) X-ray photoelectron spectroscopy (XPS) survey spectra of SeTe-CuO NPs. (E) Deconvoluted spectra of Cu 2p of SeTe-CuO NPs. (F) Fourier transform infrared spectroscopy (FTIR) spectra of CTAB, Se-Te NPs and SeTe-CuO NPs. (G) Raman spectra of Se-Te NPs and SeTe-CuO NPs.

3) In Figure 4F, 4G, 6A and 6C, the scale bar is missing.

Response: We sincerely appreciate your comments, which greatly help us improve the quality of this work. We have checked all the figures and the missing scale bars were added. The modified Figures are given below.

Figure 3. Evaluation for *In vitro* antibacterial effect of SeTe-CuO NPs. (A) Zone of inhibitions of Se-Te NPs and SeTe-CuO NPs. (B, C) Antibiofilm activity of Se-Te NPs and SeTe-CuO NPs against *E. c.* and *S. a.*. (D) Antibacterial activity using agar plate method in the absence and presence of 808 nm laser irradiation. (E) SEM images of bacteria treated with various NPs (Scale bar = 5 µm). (F) CLSM images of live/dead staining (Scale bar = 50 µm). (G) CLSM images of intracellular ROS generation (Scale bar = 100 µm).

Figure 4. Cytotoxicity and histological study of wound tissues. (A) Representative photos of hematoxylin and eosin (H&E) staining of wounds. Scale bar = 100. (B) Cytotoxicity assessment of various NPs using CCK-8 assay. (C) Giemsa staining of wound tissues. (D) Representative images and quantitative results of the bacteria from wound of different treatment groups.

4) The photothermal conversion efficiency for SeTe-CuO NPs should be calculated.

Response: Thanks for your valuable comment. To obtain the photothermal conversion efficiency (η) for SeTe-CuO NPs. The heating and cooling curves were monitored. Thus, η was calculated referring to the following Equation (1):

$$\eta = \frac{hs(T_{max} - T_{surr}) - Q_{diss}}{I(1 - 10^{-A_{\lambda}})} \dots\dots\dots(1)$$

Where h is the heat transfer coefficient, S is the surface area of the used container, T_{max} and T_{surr} denote the maximum steady-state temperature and room temperature of the ambient environment, Q_{Diss} is the heat wastage from the light loss of the solvent and container, I is the laser intensity, and A_{λ} represents the absorbance of corresponding NPs dispersion at 808 nm. hS was calculated referring to the following Equation (2):

$$= \frac{mC_p}{hs} \dots\dots\dots(2)$$

Where τ_s is the time constant, m is the mass of the solution, and C_p is the heat capacity of corresponding solvent. Finally, the photothermal conversion efficiency can be calculated from the method above and it turns out to be 26.86%.

Figure 5. Linear time data versus $-\ln(\theta)$ obtained from the cooling period.

5) The language of this work should be polished.

Response: Thank you for your valuable suggestion. The language of the entire manuscript has been polished.

Reviewer #2:

In this manuscript, the authors synthesized SeTe-CuO NPs with dual photodynamic and photothermal properties. It revealed robust antibacterial activity and significant biofilm inhibition against pathogenic bacteria upon near-infrared (NIR) irradiation. Moreover, SeTe-CuO NPs exhibited rapid bacterial clearance within wounds, offering a promising solution for advanced wound care. Overall, the experiments were well-designed and the results seem solid. I recommend a major revision before acceptance for the publication.

Comments:

1. What is the role of SeTe in SeTe-CuO NPs?

Response: Thank you for your valuable comment. SeTe in SeTe-CuO nanoparticles plays a pivotal role in enhancing their antibacterial efficacy by synergistically improving their properties. The presence of SeTe alloys alters the optical absorption characteristics, which can be exploited in various applications, while also contributing to the generation of reactive oxygen species (ROS), a key factor in the antibacterial action of CuO. Additionally, SeTe improves the stability and dispersion of the nanoparticles, increasing the effective surface area for microbial interaction, and modulate the release of Cu ions, which are beneficial for killing bacteria. This multifunctional contribution positions SeTe-CuO NPs as a promising candidate for antimicrobial and wound healing applications.

2) In cytotoxicity investigation, it couldn't give the conclusion of "SeTe-CuO NPs were found to have good biocompatibility". The paper only discussed the assessment of SeTe-CuO NPs on live/dead bacteria, the evaluation of SeTe-CuO NPs on normal cells and other biocompatibility

experiments also need to be discussed.

Response: Thank you for your valuable suggestion. The cytotoxicity of SeTe-CuO NPs was checked at concentrations of 10, 25, 50, and 100 $\mu\text{g}/\text{mL}$ with L929 mouse fibroblast cells as given in Figure below. After 24 h incubation, the cytotoxicity of NPs was tested using the CCK-8 cell viability assay kit. As demonstrated in Figure below, with the increasing concentration of NPs, there was a negligible influence on the fibroblast cell line, L929 cell viability. Even when the concentration increased to 100 $\mu\text{g}/\text{mL}$, the cell viability remained over 95%, indicating excellent biocompatibility of the NPs.

Figure 6. Cytotoxicity evaluation of SeTe-CuO NPs

3) Compared to other CuO-based materials, what are the advantages of SeTe-CuO NPs in terms of biofilm control and antibacterial efficacy?

Response: Thanks for your valuable comment. SeTe-CuO NPs have shown promise in biofilm control and antibacterial efficacy due to their unique properties. The combination of selenium and tellurium with copper oxide can potentially enhance the antimicrobial activity of the nanoparticles. It creates a synergistic effect against bacteria. The presence of SeTe may catalyze additional redox reactions, leading to higher ROS levels, which are more effective in penetrating and disrupting biofilms. Furthermore, the photothermal activity of the NPs which reaches to a maximum of about 60°C , which is useful in eradicating biofilms. Due to having photodynamic and photothermal properties, this synergy can be particularly effective against robust biofilms, which are inherently more resistant to treatments. These properties make SeTe-CuO NPs particularly suitable for biofilm control over other CuO based materials.

In a previous report (1), researchers synthesized Ts₂CuONPs showed the highest inhibition value with $57.6 \pm 1.03\%$ at 100 $\mu\text{g}/\text{mL}$ concentration against biofilms formed by *S. aureus*, as shown in the **Figure 7**. While in our research work, the synthesized SeTe-CuO NPs at 100 $\mu\text{g}/\text{mL}$ concentration there was about an 80% reduction in biofilm for *E. coli* while about 70% of the biofilm

was inhibited in *S. aureus*.

Another study (2) also evaluated antibiofilm activity at different concentrations of synthesized NPs as given in the **Figure 8**, which is much lower than the activity of our synthesized material.

Figure 7. Inhibition effect of biosynthesized copper oxide nanoparticles on biofilm formation of *S. aureus*.

Figure 8. Biofilm survival assay of biofilm producing bacteria. (a) *A. baumannii*, (b) *S. aureus*, and (c) *K. pneumoniae*.

References

- Erci, F., Cakir-Koc, R., Yontem, M., & Torlak, E. (2020). Synthesis of biologically active copper oxide nanoparticles as promising novel antibacterial-antibiofilm agents. *Preparative Biochemistry & Biotechnology*, 50(6), 538–548. <https://doi.org/10.1080/10826068.2019.1711393>

2. Bai, B., Saranya, S., Dheepasri, V., Muniyasamy, S., Alharbi, N. S., Selvaraj, B., ... & Gnanamangai, B. M. (2022). Biosynthesized copper oxide nanoparticles (CuO NPs) enhances the anti-biofilm efficacy against *K. pneumoniae* and *S. aureus*. *Journal of King Saud University-Science*, 34(6), 102120.

4) In Legend 4, the scale in Fig.4E should be added and there is no caption for Fig.4G.

Response: Thanks for your comment. The scale bar is added in Fig. 4E and the caption for Fig. 4G is corrected as mentioned follow.

Figure 9. Evaluation for *In vitro* antibacterial effect of SeTe-CuO NPs. (A) Zone of inhibitions of Se-Te NPs and SeTe-CuO NPs. (B, C) Antibiofilm activity of Se-Te NPs and SeTe-CuO NPs against *E. c.* and *S. a.*. (D) Antibacterial activity using agar plate method in the absence and presence of 808 nm laser irradiation. (E) SEM images of bacteria treated with various NPs (Scale bar = 5 µm). (F) CLSM images of live/dead staining (Scale bar = 50 µm). (G) CLSM images of intracellular ROS generation (Scale bar = 100 µm).

REVIEWERS' COMMENTS:

Reviewer #1 (Remarks to the Author):

I am confident in stating that all of my concerns have been successfully addressed. I fully endorse the publication of the paper in Communications Chemistry.

Reviewer #2 (Remarks to the Author):

The comments have been addressed. Suggest to accept the manuscript for publication.